# SARS-CoV-2 Neutralizing Antibodies in Three African Countries Following Multiple Distinct Immune Challenges

**DOI:** 10.3390/vaccines12040363

**Published:** 2024-03-27

**Authors:** Diary Juliannie Ny Mioramalala, Rila Ratovoson, Paul Alain Tagnouokam-Ngoupo, Hermine Abessolo Abessolo, Joseph Marie Mindimi Nkodo, Georges Bouting Mayaka, Pierre Claude Tsoungui Atangana, Fanirisoa Randrianarisaona, Pulchérie Pélembi, Romaric Nzoumbou-Boko, Cathy Sandra Goimelle Coti-Reckoundji, Alexandre Manirakiza, Anjanirina Rahantamalala, Rindra Vatosoa Randremanana, Mathurin Cyrille Tejiokem, Matthieu Schoenhals

**Affiliations:** 1Institut Pasteur of Madagascar, Immunology of Infectious Diseases, Antananarivo 101, Madagascar; diaryjuliannie@pasteur.mg (D.J.N.M.); fanirisoa@pasteur.mg (F.R.); anjanirina@pasteur.mg (A.R.); 2Institut Pasteur of Madagascar, Epidemiology and Clinical Research, Antananarivo 101, Madagascar; rila@pasteur.mg (R.R.); rrandrem@pasteur.mg (R.V.R.); 3Centre Pasteur du Cameroon, Epidemiology and Public Health, Yaoundé P.O. Box 1274, Cameroon; tagnouokam@pasteur-yaounde.org (P.A.T.-N.); tejiokem@pasteur-yaounde.org (M.C.T.); 4Hôpital Central de Yaoundé, Yaoundé P.O. Box 25625, Cameroon; drabesso@yahoo.fr; 5Hôpital Jamot de Yaoundé, Yaoundé P.O. Box 4021, Cameroon; mendimajo@yahoo.fr; 6Hôpital de District d’Obala, Obala P.O. Box 0000, Cameroon; bouting1972@gmail.com; 7Hôpital de District de Mbalmayo, Mbalmayo P.O. Box 147, Cameroon; p4043@yahoo.com; 8Institut Pasteur of Bangui, Epidemiology, Bangui P.O. Box 923, Central African Republic; pulcherie.pelembi@pasteur-bangui.cf (P.P.); romaric.nzoumbou-boko@pasteur-bangui.cf (R.N.-B.); sandra.coti@pasteur-bangui.cf (C.S.G.C.-R.); alexandre.manirakiza@pasteur-bangui.cf (A.M.)

**Keywords:** neutralizing antibodies, natural infection, vaccination, healthcare worker, COVID-19

## Abstract

Background: The COVID-19 pandemic has affected Madagascar, Cameroon, and the Central African Republic (CAR), with each experiencing multiple waves by mid-2022. This study aimed to evaluate immunity against SARS-CoV-2 strains Wuhan (W) and BA.2 (BA.2) among healthcare workers (HCWs) in these countries, focusing on vaccination and natural infection effects. Methods: HCWs’ serum samples were analyzed for neutralizing antibodies (nAbs) against W and BA.2 variants, with statistical analyses comparing responses between countries and vaccination statuses. Results: Madagascar showed significantly higher nAb titers against both strains compared to CAR and Cameroon. Vaccination notably increased nAb levels against W by 2.6-fold in CAR and 1.8-fold in Madagascar, and against BA.2 by 1.6-fold in Madagascar and 1.5-fold in CAR. However, in Cameroon, there was no significant difference in nAb levels between vaccinated and unvaccinated groups. Conclusion: This study highlights the complex relationship between natural and vaccine-induced immunity, emphasizing the importance of assessing immunity in regions with varied epidemic experiences and low vaccination rates.

## 1. Introduction

The severe acute respiratory syndrome coronavirus 2 (SARS-CoV-2) has given rise to a pandemic, as declared on 11 March 2020 by the World Health Organization [1]. In the African continent, the first reported case of COVID-19 occurred in Egypt on 14 February 2020 [2]. Despite Africa experiencing comparatively lower rates of infection and mortality throughout the pandemic, the reported numbers may not accurately reflect the true extent of the situation due to limited detection capabilities [3]. Various SARS-CoV-2 variants have emerged, with only certain ones classified as variants of concern (VOCs) based on their significant impact on public health. VOCs are associated with particular characteristics, such as reduced neutralization by antibodies acquired through prior infection or vaccination. Four major waves had been associated mainly with the ancestral Wuhan strain, Alpha, Beta, Delta, and Omicron VOCs, in Africa at the time of this study. These epidemic waves successively appeared without a large inter-peak and ended respectively in September 2020, April 2021, October 2021, and March 2022 [4].

Studying landscape immunity in South Africa in populations infected at least once by SARS-CoV-2 revealed that prior infection can confer protection against newly circulating viruses [5]. In April 2022, Madagascar and the Central Africa Republic (CAR) experienced three epidemic waves with comparable overall profiles, while Cameroon had undergone four distinct epidemic waves [4]. In Madagascar, the second epidemic wave was associated with the circulation of the VOC Beta (GH/501Y.V2), as previously reported by GISAID Dashboard [6]. The third bimodal wave was the result of a combination of Delta and Omicron (B.1.1.529) VOCs circulation, with apparently smaller peaks [7]. On the other hand, CAR was hit by VOC Alpha during the second wave, followed by a smaller bimodal wave, as in Madagascar with the co-circulation of Delta and Omicron during the third wave [4]. For Cameroon, the second wave was caused by the VOC Beta (GH/501Y.V2), as in Madagascar, but followed by two independent waves corresponding to the circulation of Delta and then Omicron [6]. Figure 1 illustrates the epidemic trends of reported cases and detected variants during these waves.

A previous study has shown that infection-derived immunity can vary depending on the infecting variant [8]. Furthermore, individuals who were infected with the VOC Beta experienced more severe outcomes compared to those infected with the Alpha variant, which exhibited higher vaccine efficacy. During the sampling period of the current study, VOC Omicron was circulating in these countries. Servellita et al. assumed that the Omicron spread could have contributed to the mass immunization in the population, potentially leading to the end of the COVID-19 pandemic. This Omicron-driven wave had reduced virulence due to prior infection and/or vaccination [9]. Moreover, Delta and Omicron infections have been shown to enhance immune responses against other variants, including the ancestral strain and other VOCs [10].

Vaccination has been shown to enhance protection conferred by a previous infection [11] in specific contexts. This immunity, known as hybrid immunity, is said to confer strong protection [12]. It has been described that infection with VOC Omicron induces cross-neutralizing antibody responses by reinforcing pre-existing immunity conferred by natural infection or vaccinations [13,14]. Consequently, measuring neutralizing antibodies can serve as an effective tool to quantitatively assess the protection conferred by natural immunity, vaccination, or a combination of both [15]. This approach can also help in predicting the level of protection against re-infection, particularly concerning potential future VOCs [16]. The population in Madagascar and Cameroon exhibited low vaccination coverage, with less than 10% vaccination coverage in April 2022, while the CAR had a vaccination coverage of 22.1% [17] at the same time.

Healthcare workers (HCWs) are considered vulnerable populations given their high risk of exposure [18]; they are also on the front line of vaccination campaigns. Indeed, the pooled estimated prevalence of HCW acceptance for the COVID-19 vaccine was 56.6% in 2022 [19]. Investigating the immunity of these highly exposed populations in different epidemiological contexts may provide valuable immunization descriptions.

Our study aimed to analyze the effectiveness of immunization among HCW populations from three different countries against a future variant. This multicentric investigation thus evaluated neutralization profiles against two strains of the SARS-CoV-2: the original Wuhan strain (W) and the Omicron BA.2 variant (BA.2), following the third African wave of COVID-19 infections in 2022.

## 2. Materials and Methods

### 2.1. Study Population and Serum Sampling

#### 2.1.1. Participants from Madagascar

From May to June 2022, a cross-sectional study was set up to estimate SARS-CoV-2 seroprevalences at the end of the third bimodal epidemic wave and targeting HCWs. This study was carried out in 3 university hospitals in Antananarivo: CHU Joseph Raseta Befelatanana, CHU Anosiala, and CHU de Soavinandriana. A standardized questionnaire was used at inclusion to collect information about previous COVID-19 infection and vaccination. For HCWs who declared vaccination, vaccination cards were requested when available. Five mL (5 mL) blood samples were collected from each participant and then stored at −20 °C.

#### 2.1.2. Participants from Central Africa Republic

In May 2022, our survey took place at three key healthcare institutions in Bangui, the capital of the Central African Republic: The University Hospital of the Sino-Central African Friendship (CHUASC), the Community University Hospital (CHUC), and the Maman Elisabeth Domitien University Hospital (CHUMED). These hospitals, namely CHUMED, CHUASC, and CHUC, are esteemed national teaching hospitals. Our study encompassed all healthcare personnel involved in patient care, including medical doctors, nurses, midwives, radiology technicians, laboratory technicians, surgeons’ staff, as well as support staff with non-clinical roles within the healthcare departments of these institutions. Demographic information such as gender and age, along with qualifications, was collected from the participating healthcare workers. Additionally, blood samples of approximately 1 to 2 mL were obtained in dry tubes. Serums were subsequently stored at −20 °C and then transferred to the Institut Pasteur de Madagascar for further analysis.

#### 2.1.3. Participants from Cameroon

From July to August 2022, a cross-sectional study was conducted among HCWs in two referral hospitals located in Yaoundé (the Jamot Hospital and the Specialized Center for the Care of COVID-19 Patients, Annex 2, Central Hospital) and two district hospitals (Obala and Mbalmayo). Any staff (health, administrative, and support) whose names were on the official staff list transmitted to our research team by hospital authorities and who consented in writing to participate on a voluntary basis in this study were included. A standardized self-questionnaire was used at inclusion to collect information about sociodemographic characteristics, previous COVID-19 infection, and vaccination. For HCWs who declared themselves to be vaccinated, vaccination cards had been requested if available. Five milliliters (5 mL) of blood samples were collected from each participant, centrifuged, and plasma stored at −80 °C in Centre Pasteur du Cameroun. Plasma samples were subsequently transferred to the Institut Pasteur de Madagascar for further analysis.

### 2.2. Pseudovirus Neutralizing Assay

The pseudovirus neutralizing assay is a test to measure the neutralization capacity of antibodies present in the serum of each individual by simulating the presence of the virus using a pseudovirus. Neutralizing antibodies against SARS-CoV-2 ancestral strain (W) and SARS-CoV-2 Omicron BA.2 (BA.2) were measured using SARS-CoV-2 luciferase reporter virus particles (RVP-701L and RVP-770L Integral molecular^®^, Philadelphia, PA, USA), following the manufacturer’s protocol. Briefly, sera were diluted from 1:10 to 1:5120; pseudoviruses were added and pre-incubated at 37 °C for one hour. Cells expressing the receptor of the pathogen, here “293T-hsACE2” (C-HA102 Integral molecular^®^, Philadelphia, PA, USA), were added to each well and then incubated for 72 h in a 5% CO_2_ environment at 37 °C. The luminescence (RLU) of the cells was measured on a Microplate Luminometer (Luminoskan™, Waltham, MA, USA) using a Renilla-Glo Luciferase assay system (E2710 Promega^®^, Fitchburg, WI, USA). RLU was normalized as follows:(1)$$\frac{\left[\left(RLU\;virus\;control\;well-RLU\;Cell\;control\;well\right)-\left(RLU\left(Serum+virus\right)-RLU\;Cell\;control\;well\right)\right]}{\left(RLU\;Virus\;control\;well-RLU\;Cell\;control\;well\right)}\;\times \;100$$

The neutralization titer was determined by analyzing luminescence values and determining the dilution point at which 50% infectivity occurred through a four-parameter regression.

### 2.3. Data Analysis

Data analysis was conducted using GraphPad Prism version 8.0 (GraphPad Software Inc., La Jolla, CA, USA). Neutralizing antibodies were reported as medians with ranges, and Mann–Whitney tests were used to compare titers among different groups (Madagascar, Cameroon, and CAR/Unvaccinated and Vaccinated). Fischer’s exact tests were employed to compare percentages. All tests were considered statistically significant at a *p*-value < 0.05.

Positivity thresholds were established using a Gaussian mixture model, which identified the threshold by optimizing the log-likelihood function through the expectation-maximization algorithm. This method assumes multiple Gaussian underlying distributions (in our case 2) for “negative” and “positive” individuals. The Gaussian mixture model analysis was performed using the Mclust package within R-studio software version 4.2.1.

## 3. Results

### 3.1. Study Populations

In Madagascar, data collection was conducted from 13 May to 23 June 2022. A total of 558 eligible HCWs were invited to participate, and 512 (91.7%) accepted. Participants were predominantly females (59.4%). Physicians represented 21.9% of participants; medical students, 14.1%; paramedics, 35.4%; and other functions (such as a physiotherapist, lab staff, stretcher-bearer), 28.7%. The median age was 31.1 years (Interquartile range IQR = 26.2–40.9), and 31.2% reported having at least one comorbidity (commonly hypertension, cardiopathy, diabetes, auto-declared obesity, and asthma or other pulmonary chronic diseases). At inclusion, 23.8% of participants reported having respiratory symptoms. Among the participants, 64.3% had already been vaccinated at least once, 79.7% had been infected at least once, and 26.26% experienced more than one infection. A total of 17.8% were infected during the first wave of COVID-19, 34.2% in the second wave, and 34% during the third wave. Among multi-infected individuals, 12.7% of individuals reported being re-infected once, while 1.3% reported undergoing reinfection two times.

In CAR, samples were collected in May 2022. One hundred forty-one HCWs were included. The majority were female (63.1%). Among the study participants, physicians comprised 4.2%, while medical students constituted 1.4%, paramedics accounted for 55.3%, and individuals with various roles constituted 39.1%. The median age of the cohort was 43 years (IQR = 26.2–40.9). Additionally, 21.6% of participants reported the presence of at least one comorbidity, most commonly hypertension, cardiopathy, diabetes, self-reported obesity, asthma, or other chronic pulmonary diseases.

In Cameroon, data collection was conducted from July to August 2022. Samples of 200 HCWs identified during this period were sent to Madagascar. Among these HCWs, physicians comprised 7.0%, while medical students constituted 1.0%, paramedics accounted for 58.0%, and individuals with various roles constituted 34.0%. The median age of the cohort was 34.0 years (IQR = 29.0–39.5). Additionally, 15.0% of HCWs reported the presence of at least one comorbidity, and 61.5% received at least one dose of COVID-19 vaccine, confirmed in 91.9% with presentation of a valid vaccination card. Table 1 shows the detailed sociodemographic characteristics of the participants.

### 3.2. Madagascar HCW nAb Levels Were Higher than Those from CAR and Cameroon

Serological analysis was performed to define total IgG anti-S1, anti-RBD, and anti-NP seropositivities. Cut-off limits for the determination of positive individuals are described previously [20]. In Madagascar, 95% (95%CI = 93–97), 92% (95%CI = 90–95), and 75% (95%CI = 67–75) of the cohort tested positive for anti-RBD, anti-S1, and anti-NP antibodies, respectively. Anti-S1, anti-RBD, and anti-NP antibody seroprevalences were 97.84% (95%CI = 95.40–100), 97.84% (95%CI = 95.40–100), and 69.06% (95%CI = 61.28–76.84), respectively, in RCA. Similarly, in Cameroon, the cohort exhibited seropositivity rates of 89% (95%CI = 84.63–93.37), 87% (95%CI = 82.3–91.7), and 60% (95%CI = 57.3–66.07) for anti-S1, anti-RBD, and anti-NP antibodies, respectively (Figure 2).

To assess the level of protection against the ancestral W strain and against the BA.2 VoC among HCWs in Madagascar, CAR, and Cameroon, we compared the corresponding neutralizing antibody titers (nAbs) in these countries. The findings indicated that the percentage of nAbs against the ancestral strain W was notably higher in Madagascar (Madagascar vs. CAR, *p* = 0.03; Madagascar vs. Cameroon, *p* < 0.01), while it remained consistent in CAR and Cameroon (*p* = 0.20) (Figure 3a). Similar trends were observed in nAb titers against the BA.2 variant. Madagascar exhibited a 2.2-fold higher nAb titer in anti-BA.2 compared to Cameroon (*p* < 0.01) and a 1.6-fold higher one than in CAR (*p* < 0.01). Conversely, nAb titers were comparable between CAR and Cameroon (*p* = 0.12) (Figure 3b).

We then examined the rate of individuals positive by calculating the frequency of individuals with antibodies above the defined threshold (nAb titers cut-off W = 719.42 and BA.2 = 329.7). There was no difference in terms of neutralization against W (Figure 3c). In line with the nAb results, the BA.2 nAb positivity rate was significantly higher in Madagascar compared to CAR (31.51%, 95% CI: 27.47–35.55 vs. 18.57%, 95% CI: 12.05–25.09, *p* = 0.01) (Figure 3d). Cameroon HCWs (27.00%, 95% CI: 20.79–33.21) had nAb positivity rates similar to those of Madagascar and CAR ones.

### 3.3. Vaccination Increased nAb Levels in Madagascar and CAR but Not in Cameroon

We then analyzed nAb levels in differentially vaccinated groups. We found a 2.6-fold higher (*p* < 0.01) nAb level against W pseudotyped particles in the samples from vaccinated CAR individuals vs. their unvaccinated colleagues and a 1.8-fold higher (*p* < 0.01) nAb level in vaccinated Madagascar individuals vs. their unvaccinated peers (Figure 4a). Additionally, vaccination appeared to boost BA.2 variant nAb titers in HCW populations by 1.6-fold in Madagascar (*p* < 0.01) and 1.5-fold in the CAR region (*p* = 0.04) (Figure 4b). Interestingly, Cameroon showed no difference in nAb titers against either W or BA.2 (Figure 4a,b).

A comparative analysis of nAb titers against the W variant of SARS-CoV-2 within unvaccinated and vaccinated cohorts across these countries showed that the unvaccinated group in CAR exhibited nAb titers that were two times lower than those observed in Madagascar (*p* < 0.01) and Cameroon (*p* = 0.04) (Figure 4a). Interestingly, Madagascar and Cameroon demonstrated comparable nAb titers against BA.2 (Figure 4b). Conversely, within the vaccinated groups, Madagascar HCWs did not exhibit different nAb titers (*p* = 0.06) from those from CAR, despite a lower vaccination rate in Madagascar (64.7% vs. 84.2%) (Figure 4a). In Cameroon-vaccinated HCW samples, nAb titers were 2-fold lower than in Madagascar (*p* < 0.01) and 1.8-fold lower than in CAR (*p* < 0.01). However, HCWs in Madagascar always showed stronger neutralization against BA.2 in both the unvaccinated (*p* < 0.01 and *p* = 0.02) and the vaccinated group (*p* < 0.01 and *p* < 0.01) compared to both CAR and Cameroon. In contrast, CAR and Cameroon samples showed no nAb titer differences in the unvaccinated group (*p* = 0.08) and vaccinated group (*p* = 0.61) (Figure 4b).

Interestingly, the positivity rate of the unvaccinated group against W in Cameroon was higher than in CAR (*p* < 0.01) and Madagascar (*p* < 0.01), but no significant difference was observed concerning the vaccinated group. The same trend was observed for BA.2 nAbs, even though this did not reach significance in the unvaccinated group (Figure 4c,d).

### 3.4. Vaccination and Infection Provided Similar BA.2 Neutralization in Madagascar

To further investigate the implications of different types of immunization, participants from Madagascar were classified into 4 groups: 38 unvaccinated/uninfected (I−V−), 146 unvaccinated/infected (I+V−), 65 vaccinated/uninfected (I−V+), and 262 vaccinated/infected (I+V+). In the I+V+ group, W strain neutralization was 2.3 times higher than in the I−V− group (*p* < 0.01) and 1.8 times higher than in the I+V− group (*p* < 0.01) (Figure 5a). Moreover, infected groups appeared to have a higher positivity rate independently of previous vaccination (I−V− vs. I+V−: *p* < 0.01, I−V− vs. I−V+: *p* = 0.55, I−V− vs. I+V+: *p* = 0.14; I+V− vs. I−V+: *p* < 0.01, I+V− vs. I+V+: *p* = 0.02, I−V+ vs. I+V+: *p* < 0.50) (Figure 5c).

On the other hand, nAb levels against BA.2 were similar for all analyzed groups (Figure 5b). However, consistent with the response against W, the nAb positivity rate in the surveyed infected groups was higher regardless of vaccination status (I−V− vs. I+V−: *p* = 0.03, I−V− vs. I−V+: *p* = 0.06, I−V− vs. I+V+: *p* = 0.18; I+V− vs. I−V+: *p* < 0.01, I+V− vs. I+V+: *p* = 0.57, I−V+ vs. I+V+: *p* < 0.01) (Figure 5d).

Among unvaccinated individuals who declared having been infected only once, 3.45% reported infection during the 1st wave, 13.95% reported infection during the 2nd wave, and 27.9% reported infection during the 3rd wave (Appendix A). BA.2 nAb titers tended to gradually decrease over time since the last reported infection. Indeed, individuals infected only during the first wave seemed to have a lower response than those infected during the second or third wave, although the differences did not reach significance due to the small size of the analyzed group (Appendix A).

## 4. Discussion

Healthcare workers (HCWs) are at an elevated risk of SARS-CoV-2 infection compared to the general population, due to their frequent exposure to contagious individuals. The implementation of appropriate infection prevention and control programs is crucial to safeguard HCWs. However, the availability of collective and personal protective equipment remains limited, especially in Africa. Even low-cost interventions such as facemasks and a water supply for handwashing may be challenging in this particular setting [21]. The scope of this study focused on HCWs in three African countries: Madagascar, Cameroon, and the Central African Republic (CAR). Remarkably, these countries exhibited nearly identical seroprevalence rates, with 95% (Madagascar) [7], 98.4% (Cameroon) [22], and 95.7% (CAR) [23], respectively. These high seroprevalences indicate that a substantial proportion of HCWs have been exposed to the virus, making these populations valuable targets for comparative immunization studies. Notably, HCWs represent an important group for monitoring and evaluating infection and immunity trends in Africa, including the assessment of vaccination efficacy. As this group plays a central role in the healthcare system, understanding their immunization status can shed light on the effectiveness of vaccination efforts in the region.

At the beginning of this study, vaccination rates in the study population in Madagascar, Cameroon, and CAR were, respectively, 64.3%, 58.0%, and 82.2%. The vaccination rate among HCWs in Madagascar was comparable to that reported in HCWs across Africa (65.6%) but lower than the global average (77.3%) [24]. By April 2022, the World Health Organization (WHO) declared that 65% of Africans had already been infected by BA.2 subvariant. The current study thus represents the first report of HCW protection against VOC BA.2 profiles (neutralization potential) in Madagascar, Cameroon, and the CAR, before it being detected in these countries [6]. The level of protection was assessed by measuring the neutralizing antibodies, as previous research has demonstrated that a high level of neutralizing antibodies could serve as a good predictor of protection against COVID-19 [25]. Our results revealed a similar level of neutralizing antibodies (nAbs) against the Wuhan strain (W) in both Cameroon and CAR, while it was higher in Madagascar. Prior studies have demonstrated the ability of our immune system to better eliminate this ancestral strain W following Omicron immunization [26,27]. Additionally, vaccination has been found to reinforce protection against the Wuhan strain infection. Indeed, the initial vaccines were developed from this strain [28]. In line with these observations, our results indicate that vaccinated groups exhibited the strongest response against strain W in Madagascar and CAR (Figure 4a). Interestingly, vaccinated HCWs in Madagascar showed the same level of neutralization activity as those in CAR, despite the low vaccination coverage, suggesting stronger immunization resulting from infections [7,29]. This result was confirmed by a high level of infection reported in the Malagasy HCW population (79.7%) (Table 1) and the serological result assessing IgG anti-S1, anti-RBD, and anti-NP, which revealed that Madagascar’s HCWs had higher, though not significantly, NP titers, suggesting a more recent natural infection [30] (Figure 2). Seropositivities for anti-S1 and anti-RBD IgGs in the RCA compared to Cameroon and Madagascar were also higher and could be correlated with a higher vaccine coverage in the RCA [31]. On the other hand, we found that the rate of individuals neutralizing W in the unvaccinated group in Cameroon was higher (Figure 4c,d) compared to the unvaccinated groups in CAR and Madagascar, despite only 22% having reported a previous infection. This surprising result could be due to low reporting and/or little symptomatic presentation in the context of high natural immunization (four successive waves) [4]. WHO has indeed stated that the reported number of COVID-19 infection cases in Cameroon was underestimated [32].

Focusing on HCWs in Madagascar, participants were classified based on their known past immunization history. Individuals who had both been infected and vaccinated showed higher neutralization activities of W pseudotyped particles compared to vaccinated and uninfected or unvaccinated individuals. These data are consistent with previous reports that suggest that hybrid immunity resulting from both infection and vaccination confers better protection than vaccination or infection alone [33,34,35]. Our results further support the effectiveness of vaccination in providing protection against strain W.

Subsequently, we investigated pre-existing immunization against BA.2, anticipating a future potential variant that had not yet circulated at the time of this study. HCW samples from CAR did not exhibit a response or only exhibited a weak response against BA.2, suggesting a mild pre-existing immunity. Malagasy HCWs, however, showed higher BA.2 neutralization potential. This finding can be attributed to the overall population that was heavily re-infected during the 3rd bimodal wave without particular clinical symptom presentation, in line with little triple infection reporting [36]. This may as well be associated with an efficient cellular response to SARS-CoV-2 having led to fewer symptomatic presentations [7]. Moreover, Omicron infections have been generally reported to be asymptomatic or less severe [37]. Cameroon samples from unvaccinated and vaccinated groups exhibited similar BA.2 nAbs levels (Figure 4a,b). Moreover, non-infected unvaccinated individuals also showed high nAb levels (Appendix A), indicating possible under-reporting during the third wave in this country.

The current study, however, has limitations that should be acknowledged. All collected information was not collected uniformly across all countries, preventing sub-populational analysis in all countries. Previous infections were self-reported only. Additionally, vaccination strategies, including details such as the vaccine manufacturers and the number of vaccine doses received, were not included in the analysis of all countries.

Healthcare workers not only constitute a highly exposed population but also serve as indicators of the effectiveness of the implemented protection strategies. Our study aimed to assess whether immunization among healthcare workers in Madagascar, Cameroon, and the Central African Republic enabled these populations to effectively combat future variants. Our findings suggest that neutralization varies in populations presenting similar immunization schemes due to differences in strains and probably distance between strains. These results emphasize the need to better describe the intricate interplay between natural and vaccine-induced immunities in regions characterized by diverse strain dynamics which have significantly influenced epidemiological landscapes.

## Figures and Tables

**Figure 1 vaccines-12-00363-f001:**
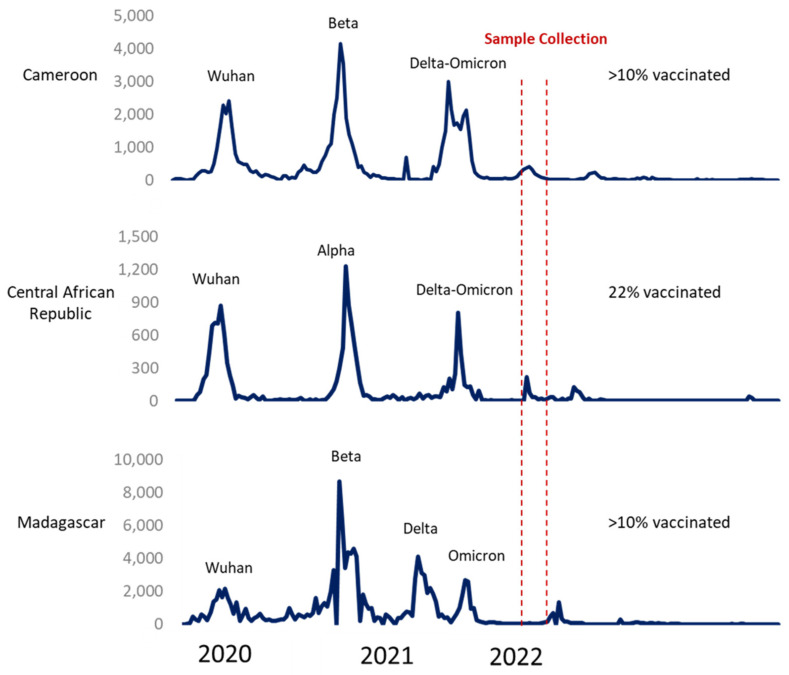
COVID-19 dashboard: Cameroon, Republic of Central Africa, and Madagascar. Number of weekly reported cases (WHO) and VOC detection (GISAID).

**Figure 2 vaccines-12-00363-f002:**
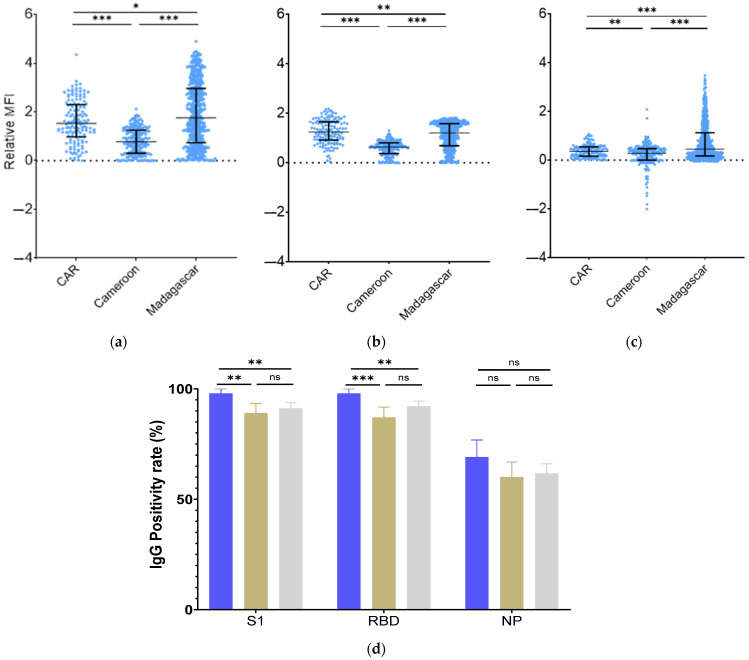
Anti-S1, anti-RBD and anti-NP Antibodies (**a**–**c**) Dotplot representing MFI levels for specific anti-S1, anti-RBD, anti-NP IgGs. MFI: Median Fluorescence Intensity. Data were compared using the Mann–Whitney test. *: *p*-value < 0.033; *** *p*-value < 0.0001. Each dot represents an independent sample. (**a**): IgG anti-S1, (**b**): IgG anti-RBD, (**c**): IgG anti-NP (**d**): Percentage of individuals having IgG titers above positivity threshold. Data were compared using the Fischer’s exact test. Blue: CAR, Brown: Cameroon, Grey: Madagascar. ** *p*-value < 0.001; *** *p*-value < 0.0001; ns: non-significative.

**Figure 3 vaccines-12-00363-f003:**
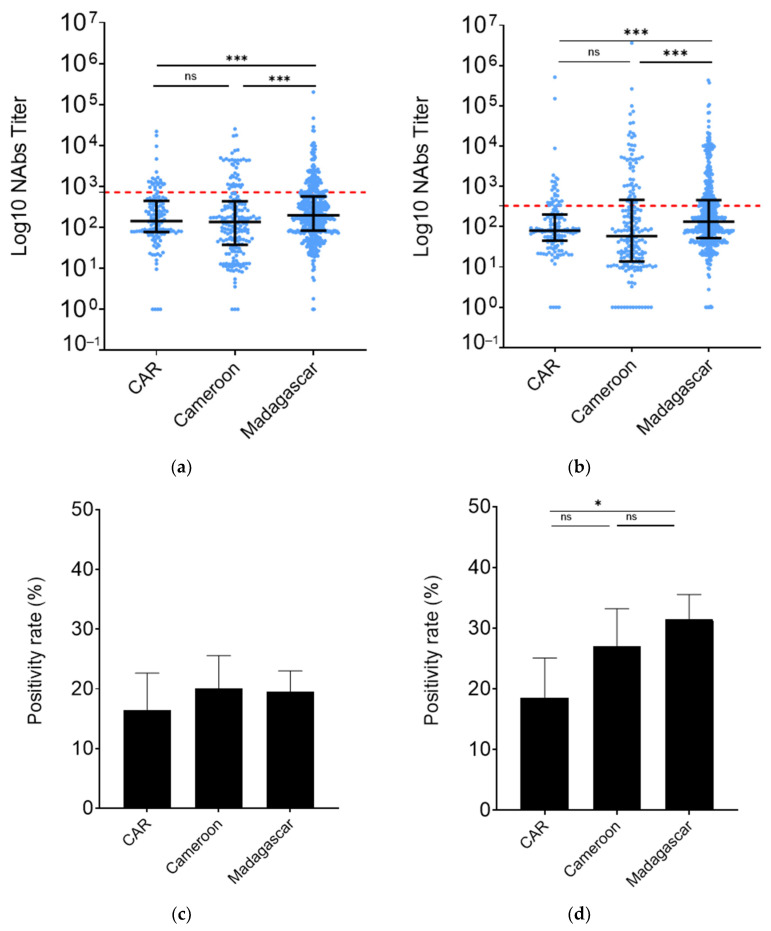
HCW nAbs in Cameroon, Republic of Central Africa, and Madagascar. (**a**) Against WT, (**b**) against BA.2. Populational groups are indicated on the X axis and neutralization titers on the Y axis. Data were compared using the Mann–Whitney test. Each dot represents an independent sample. (**c**,**d**) Percentage of individuals having nAb titers above the calculated threshold. (**c**) Against WT, (**d**) against BA.2. Data were compared using the Fischer’s exact test. * *p*-value < 0.05; *** *p*-value < 0.0001; ns: non-significative.

**Figure 4 vaccines-12-00363-f004:**
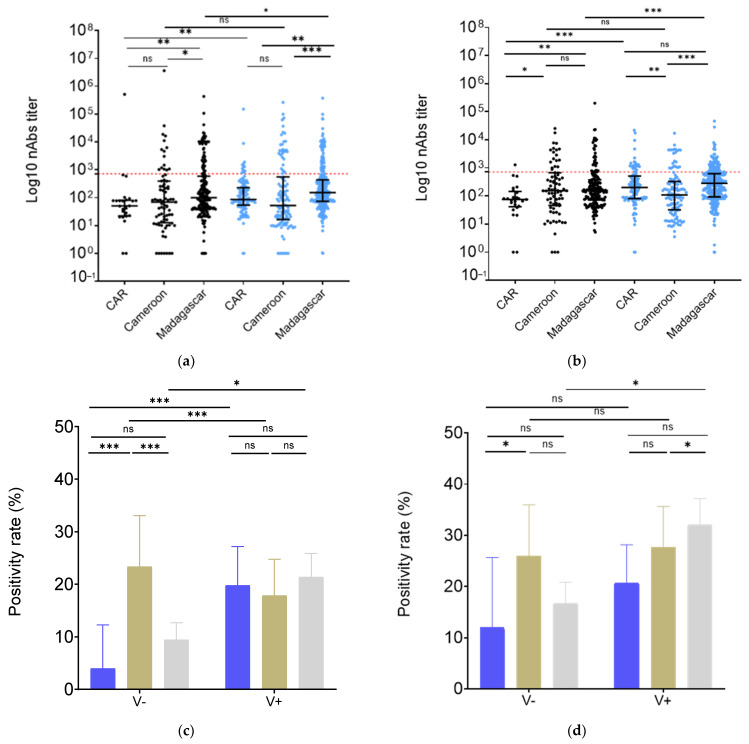
HCW nAbs in Madagascar, Cameroon, and CAR depending on vaccine status. (**a**) Against WT, (**b**) against BA.2. Populational groups are indicated on the X axis and neutralization titers on the Y axis. Data were compared using the Mann–Whitney test. Each dot represents an independent sample. Grey: unvaccinated, Blue: Vaccinated. (**c**,**d**) Percentage of individuals having nAb titers above the calculated threshold. (**c**) Against WT, (**d**) against BA.2. Data were compared using the Fischer’s exact test. Blue: CAR, Brown: Cameroon, Grey: Madagascar. * *p*-value < 0.05; ** *p*-value < 0.001; *** *p*-value < 0.0001; ns: non-significative.

**Figure 5 vaccines-12-00363-f005:**
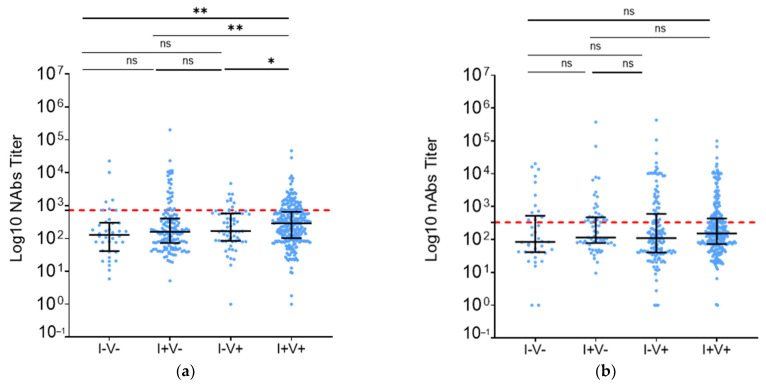
HCW nAbs in Madagascar depending on immunization status. (**a**) Against WT, (**b**) against BA.2. Populational groups are indicated on the X axis and neutralization titers on the Y axis. Data were compared using the Mann–Whitney test. Each dot represents an independent sample. (**c**,**d**) Percentage of individuals having nAb titers above the calculated threshold. (**c**) Against WT, (**d**) against BA.2. Data were compared using the Fischer’s exact test. * *p*-value < 0.05; ** *p*-value < 0.001; *** *p*-value < 0.0001; ns: non-significative.

**Table 1 vaccines-12-00363-t001:** HCW participants’ characteristics in Central Africa Republic, Cameroon, and Madagascar.

Characteristics		Madagascar HCW *n* (%)	CAR HCW *n* (%)	CMR HCW *n* (%)
Total included		512	141	200
Median age [IQR]		31.1	43	34.0
		[26.2–40.9]	[36–51]	29.0–39.5
Gender				
Female		304 (59.4)	89 (63.1)	139 (60.5)
Male		208 (40.6)	52 (36.9)	61 (30.5)
Function				
Physicians		112 (21.9)	6 (4.2)	14 (7.0)
Medical students		181 (35.4)	2 (1.4)	2 (1.0)
Paramedics (nurses, midwife)		72 (14.1)	78 (55.3)	116 (58.0)
Other		147 (28.7)	55 (39.1)	68 (34.0)
Hospitals *				
1		136 (26.6)	61 (43.2)	42 (21.0)
2		339 (66.2)	35 (24.8)	60 (30.0)
3		37 (7.2)	45 (32)	45 (22.5)
4		-	-	53 (26.5)
Comorbidities				
No		352 (68.8)	78 (56.5)	170 (85.0)
At least one		160 (31.2)	31 (21.6)	30 (15.0)
Unknown		-	32 (21.9)	-
Respiratory symptoms 15 days prior to inclusion				
No		390 (76.2)	98 (69.5)	196 (98.0)
Yes		122 (23.8)	43 (30.5)	4 (2.0)
Vaccination				
At least one dose		329 (64.3)	116 (82.2)	123 (61.5)
Not vaccinated		183 (35.7)	25 (17.8)	77 (38.5)
Vaccination card available		62 (12.1)	-	113 (91.9)
N doses received				
1 dose		173 (33.8)	37 (26.2)	-
2 doses		106 (20.7)	79 (56.1)	-
3 doses		50 (9.8)	25 (17.7)	-
Reported COVID-19 infection				
At least once		408 (79.7)	5 (3.5)	44 (22.5)
	Infected during the 1st wave (2020) **	91 (17.8)	3 (2.1)	-
	Infected during the 2nd wave (1st semester 2021) **	175 (34.2)	5 (3.5)	-
	Infected during the 3rd wave (end of 2021–1st trimester 2022) **	174 (34.0)	-	-
Never infected		103 (20.1)	136 (96.5)	156 (78.5)
Unknown		1 (0.2)	0 (0)	-

* Madagascar 1 = Anosiala; 2 = Befelatanana; 3 = Soavinandriana/CAR 1 = Sino-Central African Friendship; 2 = Community University; 3 = Maman Elisabeth Domitien/Cameroon 1 = Hôpital Jamot de Yaoundé; 2 = Centre spécialisé de Prise en charge des Patients COVID-19; 3 = Hôpital de district de Mbalmayo; 4 = Hôpital de District d’Obala. ** The percentages correspond to the number of HCWs infected during each wave. One HCW may be infected multiple times.

## Data Availability

Data will be made available upon request to the corresponding author.

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
