# Peer review of "SARS-CoV-2 Neutralizing Antibodies in Three African Countries Following Multiple Distinct Immune Challenges"

_vaccines, 2024, doi:10.3390/vaccines12040363_

Round 1

Reviewer 1 Report

Comments and Suggestions for Authors

Ny Mioramalala et al. describe their findings on SARS-CoV-2 neutralizing antibodies in three African countries  following multiple distinct immune challenges.

The study is well built, uses blood samples collected from health care workers – a population that is strongly exposed to respiratory virus infections and also with a high proportion of vaccinated individuals. Statistical evaluation is fine, illustrations are clear. Conclusions are supported by the observations, no unsupported theories are built. As the authors honestly admit at the end of the article, information on the type of vaccine used for the vaccinations could have added great value to this study.

The English of the article is good. Please, find  a few suggestions below for consideration:

Line 47 march – March

Line 106-109, 123-126, 140-142 Move these parts to the end of the article where ethical statements are collected according to the Journal’s format.

Line 150 Cells expressing the receptor of the pathogen here "293T-hsACE2" (Integral, C-HA102) was added – were added

Line 212-213 Madagascar exhibited a 2.2-fold increase in anti-BA.2 nAbs compared to Cameroon (p<0.01) and a 1.6-fold increase compared to CAR (p<0.01). – Please, reformulate. The nAb level was higher in Madagascar than in Cameroon and CAR, but I would not call it to be an “increase” compared to them. You can detect an increase within a category but not between categories.

Line 231 and a 1.8-fold increase (p<0.01) in Madagascar vaccinated individuals vs. theirs (Figure 3A). – Please, consider to add who is “theirs” for better readability. Suggestion: . 231 Madagascar vaccinated individuals vs. their non-vaccinated peers.

Comments on the Quality of English Language

Coments on a few points related to language are included among the general comments.

Author Response

The English of the article is good. Please, find a few suggestions below for consideration:

Line 47 march – March

Line 106-109, 123-126, 140-142 Move these parts to the end of the article where ethical statements are collected according to the Journal’s format.

Line 150 Cells expressing the receptor of the pathogen here "293T-hsACE2" (Integral, C-HA102) was added – were added

Line 212-213 Madagascar exhibited a 2.2-fold increase in anti-BA.2 nAbs compared to Cameroon (p<0.01) and a 1.6-fold increase compared to CAR (p<0.01). – Please, reformulate. The nAb level was higher in Madagascar than in Cameroon and CAR, but I would not call it to be an “increase” compared to them. You can detect an increase within a category but not between categories.

Line 231 and a 1.8-fold increase (p<0.01) in Madagascar vaccinated individuals vs. theirs (Figure 3A). – Please, consider to add who is “theirs” for better readability. Suggestion: 231 Madagascar vaccinated individuals vs. their non-vaccinated peers.

We would like to express our gratitude for Reviewer 1s’ thorough review of our manuscript and valuable feedback. His insightful comments have been instrumental in improving the quality of our work. We are pleased to inform him that we have carefully considered his suggestions and have made the requested amendments.

Reviewer 2 Report

Comments and Suggestions for Authors

This manuscript reports the test of neutralizing antibodies in people from 3 Africa countries. It is interesting and helpful to evaluate the population immunity against COVID19 in these countries and can reflecting that in the world.

The total level of antibodies against COVID19 in these people should be detected and presented in the manuscript.

The type of vaccines, which the people received in the manuscript, should be presented in the manuscript. And the results from persons received different COVID19 vaccines should be compared.

The meaning of this manuscript should be explained more carefully, in details.

Author Response

This manuscript reports the test of neutralizing antibodies in people from 3 Africa countries. It is interesting and helpful to evaluate the population immunity against COVID19 in these countries and can reflecting that in the world.

The total level of antibodies against COVID19 in these people should be detected and presented in the manuscript.

We thank Reviewer 2 for his comment regarding the total level of antibodies against COVID-19 in our study population. We had inappropriately integrated only in the discussion section (Figure S2), line 423. We have now introduced it in the Results section (lines 197-201), strengthening comprehensiveness. Figure S2 of our original submission has now been numbered S1.

The type of vaccines, which the people received in the manuscript, should be presented in the manuscript. And the results from persons received different COVID19 vaccines should be compared.

We understand Reviewer 2’s request regarding vaccine description. In many African countries including the 3 countries we investigated ref, vaccine availability and distribution was unpredictable and varied importantly, leading to an efficient but difficultly interpretable “mix-and-match” vaccinations in all 3 study populations. We therefore were not able to present a precise vaccine description for the study populations.

Rashedi R., et al. “COVID-19 vaccines mix-and-match: The concept, the efficacy and the doubts.” J Med Virol. 2022; 94: 1294-1299. doi:10.1002/jmv.27463

The meaning of this manuscript should be explained more carefully, in details.

We once again thank Reviewer 2 for this constructive comment. We have added a last paragraph (Line 358-366) in the Discussion section to strengthen the overall meaningfulness of our study.

Reviewer 3 Report

Comments and Suggestions for Authors

The manuscript submitted by Mioramalala et al., evaluates the levels of neutralizing antibodies directed against SARS-CoV-2 in African healthcare workers from Madagascar, Cameroon and Central African Republic. Samples were collected during May to August 2022 and results were analysed with regard to the vaccination status and infection history. Whereas vaccination resulted in an increase of nAb titers in Madagascar and CAR, no differences were observed for vaccinated and unvaccinated healthcare workers from Cameroon. In general, the antibody neutralization in Madagascar was higher compared to Cameroon and CAR. The authors discuss explanations for the differences that have been measured for the individual groups e.g. underreporting of infections due to the lack of clinical symptoms or differences in vaccination strategies among the three countries.

The following points should be addressed:

1.       Table 1: The information on “Reported COVID-19 Infection”, 1st/2nd and 3rd wave is not presented in a clear fashion. Was it possible to give multiple answers? To which numbers do the percentages correspond?

2.       Please indicate the threshold that has been used to determine the positivity rate of individuals.

3.       All samples have been measured in Madagascar. Samples from Cameroon were stored at -80°C and samples from CAR at -20°C. There is no information on the storage of samples from Madagascar. Has it been checked that different storage conditions do not influence the results?

4.       In Fig. 3c, the positivity rate for Madagascar V- seems to be lower than expected given that most infections occurred during the 2nd and 3rd wave (table 1), with a total of 408 infected and 183 unvaccinated individuals.

Author Response

  1. Table 1: The information on “Reported COVID-19 Infection”, 1st/2nd and 3rd wave is not presented in a clear fashion. Was it possible to give multiple answers? To which numbers do the percentages correspond?

We have revised Table 1 to enhance its clarity and facilitate understanding. Each infected healthcare worker (HCW) could provide multiple responses. The percentages represent the proportion of HCWs infected during each wave. Specifically, during the 1st wave, 91 out of 512 (17.8%) HCWs were infected in Madagascar, and 3 out of 141 (2.1%) in CAR. In the 2nd wave, these figures were 175 out of 512 (34.2%) for Madagascar and 5 out of 141 (3.5%) for CAR. In the 3rd wave, 174 out of 512 (34.0%) HCWs were infected in Madagascar.

  1. Please indicate the threshold that has been used to determine the positivity rate of individuals.

We utilized a Gaussian Mixture Model (GMM) to cluster the population into two groups based on the level of neutralizing antibodies. We have included the specific values associated with our calculations in the text.

  1. All samples have been measured in Madagascar. Samples from Cameroon were stored at -80°C and samples from CAR at -20°C. There is no information on the storage of samples from Madagascar. Has it been checked that different storage conditions do not influence the results?

We thank Reviewer 3 for this important comment that had not been addressed. While we did not specifically test the potential effects of different storage conditions on the results, and were not able to integrate this variable in our analysis, we would like to emphasize that all samples from Madagascar and the CAR were however stored under similar conditions (-20°C). We added it in the Methods section (Line 104-105). Samples were however shipped to Madagascar for Laboratory analysis in dry ice. We have however investigated the what could be the impact of sample storage temperatures on Pubmed. Kanji et al,.Ref have shown that the detection of anti-SARS-CoV-2 IgGs remained consistent regardless of storage conditions at -70°C, -20°C, 4°C, or room temperature, confirming previously described data Ref. Importantly, all the samples from our study underwent a single thaw before analysis.

Kanji, Jamil N., et al. “Stability of SARS-CoV-2 IgG in multiple laboratory conditions and blood sample types.” Journal of Clinical Virology 142 (2021): 104933.

Valo, E., et al. “Effect of serum sample storage temperature on metabolomic and proteomic biomarkers.” Sci Rep 12, 4571 (2022).:  10.1038/s41598-022-08429-0

Hess JR. “Conventional blood banking and blood component storage regulation: opportunities for improvement.” Blood Transfus. 2010 Jun;8 Suppl 3(Suppl 3):s9-15 : 10.2450/2010.003S. PMID: 20606757; PMCID: PMC2897192.

  1. In Fig. 3c, the positivity rate for Madagascar V- seems to be lower than expected given that most infections occurred during the 2nd and 3rd wave (table 1), with a total of 408 infected and 183 unvaccinated individuals.

We thank Review 3 for his careful review of our data. Indeed, this positivity rate may seem lower, however Figure 3c presents a positivity rate for Wuhan strain nAbs. Individuals infected during the second and third waves (VoC infections) seem to have have not developed strong cross-reactive to W nAbs. However, as you pointed out, in Figure 3d, we can observe a higher positivity rate for BA.2 nAbs. This highlights the importance of considering the specificity of variants when evaluating nAb positivity rates and immune responses.

Round 2

Reviewer 2 Report

Comments and Suggestions for Authors

The authors anwsered or explained the comments, which I raised previously.

The figure S1, which showed IgG against different antigen of COVID19 can be moved into manuscript. And the relationship between IgG against different COVID19 antigen and neutralizing antibodies against COVID19 virus, should be dicussed in the manuscript.

Although it is difficult to gethered the information of vaccines in Africa, at lease, the name of vaccines involved in this manuscript should be list. 

Author Response

We thank Reviewer 2 for his feedback. We have indeed included the Figure S1 in the main manuscript as suggested (now Figure 2) in briefly discussed these results in the discussion section (Lines 353-355). We have however not discussed correlations between seroprevalences and neutralization of VoCs. Antibody detection may be correlated to exposure, but seroneutralization may be used as an indicator of protection ref .

Reference:

Khoury DS, Cromer D, Reynaldi A, Schlub TE, Wheatley AK, Juno JA, et al. Neutralizing antibody levels are highly predictive of immune protection from symptomatic SARS-CoV-2 infection. Nat Med. 2021 Jul;27(7):1205–11.

Regarding the vaccines used in our study, we have listed the names of the vaccines involved as requested (Table S1). We appreciate Reviewer 2’s valuable input that has enhanced the quality and completeness of our manuscript.